# NEURAL FUNCTIONAL PROGRAMMING

**John K. Feser**
Massachusetts Institute of Technology
feser@csail.mit.edu

**Marc Brockschmidt, Alexander L. Gaunt, Daniel Tarlow**
Microsoft Research
{mabrocks,t-algaun,dtarlow}@microsoft.com

## ABSTRACT

We discuss a range of modeling choices that arise when constructing an end-to-end differentiable programming language suitable for learning programs from input-output examples. Taking cues from programming languages research, we study the effect of memory allocation schemes, immutable data, type systems, and built-in control-flow structures on the success rate of learning algorithms. We build a range of models leading up to a simple differentiable functional programming language. Our empirical evaluation shows that this language allows to learn far more programs than existing baselines.

## 1 INTRODUCTION

Inductive Program Synthesis (IPS), i.e., the task of learning a program from input/output examples, is a fundamental problem in computer science. It is at the core of empowering non-experts to use computers for repeated tasks, and recent advances such as the FlashFill extension of Microsoft Excel (Gulwani, 2011) have started to deliver on this promise.

A related line of research is the extension of neural network architectures with components that correspond to hardware primitives (Giles et al., 1989; Graves et al., 2014; Weston et al., 2015; Joulin & Mikolov, 2015; Grefenstette et al., 2015; Kurach et al., 2016; Kaiser & Sutskever, 2016; Reed & de Freitas, 2016; Andrychowicz & Kurach, 2016; Zaremba et al., 2016; Graves et al., 2016), enabling them to learn program-like behavior. However, these models are usually tightly coupled to the idea of a differentiable interpretation of computer hardware, as names such as Neural Turing Machine (Graves et al., 2014), Neural Random-Access Machine (Kurach et al., 2016), and Neural GPU (Kaiser & Sutskever, 2016) indicate. We observe that while such architectures form the basis modern computing, they are usually not the models that are used to program computers. Instead, decades of programming languages research have lead to ever higher programming languages that aim to make programming simpler and less error-prone. Indeed, as recent comparisons show (Gaunt et al., 2016b), program synthesis methods from the programming languages community that actively exploit such constructs, e.g. by leveraging known semantics of loops, are currently achieving considerably better results than comparable neural architectures. Still, neural IPS techniques are clearly at an advantage when extending the problem setting from simple integer input/output examples to more complex cases, such as IPS problems with perceptual data (Gaunt et al., 2016a), imprecise examples, or leveraging additional cues such as a natural language description of the desired program.

Hence, we propose to adapt features of modern high-level programming languages to the differentiable setting. In this paper, we develop an end-to-end differentiable programming language operating on integers and lists, taking cues from functional programming. In our empirical evaluation, we show the effects on learning performance of our four modeling recommendations, namely automatic memory management, the use of combinators and if-then-else constructs to structure program control flow, immutability of data, and an application of a simple type system. Our experiments show that each of these features crucially improves program learning over existing baselines.

## 2 BACKGROUND

The basic building block of functional programs is the function, and programs are built by composing functions together. In the following, we highlight some common features in functional programs before discussing how to integrate them into an end-to-end differentiable model in Sect. 3.

**Memory Management**   Most modern programming languages eschew manual memory management and pointer manipulation where possible. Instead, creation of heap objects automatically generates an appropriate pointer to fresh memory. Similarly, built-in constructs allow access to fields of objects, instead of requiring pointer arithmetic. Both of these choices move program complexity into the fixed implementation of a programming language, making it easier to write correct programs.

**Immutable Data**   Functions are expected to behave like their mathematical counterparts, avoiding mutable data and side effects. This helps programmers reason about their code, as it eliminates the possibility that a variable might be left uninitialized or accessed in an inconsistent state. Moreover, no data is ever "lost" by being overwritten or mutated.

**Types**   Expressive type systems are used to protect programmers from writing programs that will fail. Practically, a type checker is able to rule out many syntactically correct programs that are certain to fail at runtime, and thus restricts the space of valid programs. Access to types helps programmers to reason about the behavior of their code. In particular, the type system tells the programmer what kinds of data they can expect each variable to contain.

**Structured Control Flow**   A key difference between hardware-level assembly languages and higher-level programming languages is that higher-level languages structure control flow using loops, conditional statements, and procedures, as raw `goto`s are famously considered harmful (Dijkstra, 1968). Functional languages go a step further and leverage higher-order functions to abstract over common control flow patterns such as iteration over a recursive data structure. In an imperative language, such specialized control flow is often repeated and mixed with other code.

## 3   OUR MODELS

In the following, we will discuss a range of models, starting with a simple assembly-like language and progressing to a differentiable version of a simple functional programming language. We make four modeling recommendations whose effect we demonstrate in our experiments in Sect. 4.

We first discuss the general format of our programs and program states, which we will refine step by step. Our programs operate on states consisting of an instruction pointer indicating the next instruction to execute, a number of registers holding inputs and intermediate results of executed instructions, and a heap containing memory allocated by the program. We focus on list-manipulating programs, so we create a heap consisting of standard cons cells, which are data and pointer value pairs where the pointer points to another cons cell or the special `nil` value. To represent a linked list, each cell points to the next cell in the list, except for the last cell, which points to `nil`.

### 3.1   PROGRAM AND DATA REPRESENTATION

We define our models by lifting simple instructions to the differentiable setting. To do so, we bound the domain of all values and parameters, following earlier work (e.g. (Graves et al., 2014; Kurach et al., 2016; Gaunt et al., 2016b)). We represent a value $v$ from a domain $\{d_1 \ldots d_D\}$ as a tuple $\mathbb{R}^D$, interpreted as a discrete probability distribution. We pick a maximal integer value $M$ that bounds all values occurring in our programs, a number of instructions $I$, and a number of registers $R$. In this setting, the size of the heap memory $H$ has to be equal to the maximal integer value $M$, but we will relax this later. We limit the length of programs to some value $P$, and can then encode programs as a sequence of tuples $(o^{(p)}, i^{(p)}, a_1^{(p)}, a_2^{(p)})$, where $i^{(p)} \in [1, I]$ identifies the $p$-th instruction and $o^{(p)}, a_1^{(p)}, a_2^{(p)} \in [1, R]$ its output and argument registers respectively. To "execute" such a program, we unroll it for $T$ timesteps and keep a program state $s^{(t)} = (p^{(t)}, r_1^{(t)} \ldots r_R^{(t)}, h_1^{(t)} \ldots h_H^{(t)})$ for each timestep $t$, where $p^{(t)} \in [1, P]$ is an instruction pointer indicating which instruction to execute next, $r_*^{(t)}$ are the values of registers, and $h_*^{(t)}$ are the values of the cons cells in the heap.

All of our models share a basic instruction set, namely the cons cell constructor `cons`, the heap accessors (`car` & `cdr`) which return the data (resp. pointer) element of a cons cell, integer addition, increment and decrement (`add`, `inc`, `dec`), integer equality and greater-than comparison (`eq` & `gt`), Boolean conjunction and disjunction (`and` & `or`), common constants (`zero` & `one`), and finally a

`noop` instruction. These all have the usual semantics as transformers on the program state, and we will discuss the behavior of `cons` in detail later. For example, executing $(o,\ \text{add},\ a_1,\ a_2)$ on a state at timestep $t$ yields the following registers at the next timestep, where the addition operation is lifted to operate on distributions over natural numbers.

$$r_u^{(t+1)} = \begin{cases} r_{a_1}^{(t)} + r_{a_2}^{(t)} \mod M & \text{if } u = o \\ r_u^{(t)} & \text{otherwise.} \end{cases} \qquad \forall u \in [1, R]$$

As we allow all involved quantities to be distributions over all possible choices, computing the next state requires a case analysis for all allowed values. The new state is then obtained by averaging the results of all possible execution steps, weighted by the probabilities assigned to each choice. Thus, if $\eta(s^{(t)}, (o, i, a_1, a_2))$ computes the state obtained by executing the instruction $(o, i, a_1, a_2)$, we can compute the next state $s^{(t+1)}$ as follows, where $[\![x = n]\!]$ denotes the probability that a variable $x$ encoding a discrete probability distribution assigns to the value $n$.

$$s^{(t+1)} = \sum_{\substack{p \in [1,P], i \in [1,I], \\ o, a_1, a_2 \in [1,R]}} [\![p^{(t)}\!=\!p]\!] \cdot [\![o^{(p)}\!=\!o]\!] \cdot [\![i^{(p)}\!=\!i]\!] \cdot [\![a_1^{(p)}\!=\!a_1]\!] \cdot [\![a_2^{(p)}\!=\!a_2]\!] \cdot \eta(s^{(t)}, (o, i, a_1, a_2)) \quad (1)$$

In practice, we developed our models in TerpreT (Gaunt et al., 2016b), which hides these technicalities.

**Training Objective**   Our aim is to learn the program parameters $(o^{(p)}, i^{(p)}, a_1^{(p)}, a_2^{(p)})$ such that program "evaluation" according to (1) starting on a state $s^{(0)}$ initialized to an example input yields the target output in $s^{(T)}$. For scalar outputs such as a sum of values, our objective is simply to minimize the cross-entropy between the distribution in the *output* register $r_R^{(T)}$ and a point distribution with all probability mass on the correct output value.

Handling list outputs is more complex, as many valid outputs exist (depending on how list elements are placed in the heap memory). Intuitively, we traverse the heap from the returned heap address until reaching the end of a linked list, recording the list elements as we go. To formalize this intuition, let ${}^d h_k^{(T)}$ (resp. ${}^p h_k^{(T)}$) denote the data (resp. pointer) information in the heap cell at address $k$ at the final state of the evaluation. We then compute the traversal sequences of list element values $v_1, \ldots, v_H$ and addresses $a_1, \ldots, a_H$ as follows.

$$a_i = \begin{cases} r_R^{(T)} & \text{if } i = 1 \\ \sum_{a \in [1,H]} [\![a_{i-1} = a]\!] \cdot {}^p h_a^{(T)} & \text{otherwise} \end{cases} \qquad v_i = \sum_{a \in [1,H]} [\![a_i = a]\!] \cdot {}^d h_a^{(T)}$$

The probability that the computed output list is equal to an expected output list $[\bar{v}_1, \ldots, \bar{v}_k]$ is then $[\![a_{k+1} = 0]\!] \cdot \sum_{i=1}^{k} [\![v_i = \bar{v}_i]\!]$.

**Memory Management**   As the programs we want to learn need to construct new lists, we need a memory allocation mechanism that provides fresh cells. We explored two options for this allocator.

First, we attempt to follow stack-allocation models in which a stack of memory cells is used with a stack pointer $sp$ which always points to the next free memory cell. We fix a maximum stack size $H$. Whenever a memory cell is allocated (i.e., a `cons` instruction is executed), the stack pointer is incremented, guaranteeing that no cell is ever overwritten. However, uncertainty about whether an instruction is `cons` translates into uncertainty about the precise value of the stack pointer, as each call to `cons` changes $sp$. This uncertainty causes cells holding results from different instructions in the stack to blur together, despite the fact that cells are immutable once created. As an example, consider the execution of two instructions, where the first is `cons 1 0` with probability 0.5 and `noop` otherwise, and the second is `cons 2 0` with probability 0.5 and `noop` otherwise. After executing starting with $sp = 1$ and an empty stack, the value of $sp$ will be blurred across three values 1, 2 and 3 with probabilities 0.25, 0.5 and 0.25. Similarly, the value of the first heap cell will be 0 (the default) with probability 0.25, 1 with probability 0.5 and 2 with probability 0.25. This blurring effect becomes stronger with longer programs, and we found that it substantially impacted learning.

Both of these problems can be solved by transitioning to a fully immutable representation of the heap. In this variant, we allocate and initialize one heap cell per timestep, i.e., we set $H = T$. If the

current instruction is a `cons`, the appropriate values are filled in, otherwise both data and pointer value are set to a default value (in our case, $0$). This eliminates the issue of blurring between outputs of different instructions. The values of a cons cell may still be uncertain as they inherit uncertainty about the executed instructions and the values of arguments, but depend only on the operations at one timestep. While this modification requires a larger domain to store pointers, we found not copying the stack significantly reduces memory usage during training of our models.

**Recommendation (F):** Use fixed heap memory allocation deterministically controlled by the model.

## 3.2 PROGRAM MODELS

Our baseline program model corresponds closely to an assembly language as used in earlier work (Bunel et al., 2016), resulting in a program model as shown on the right, where boxes correspond to learnable parameters. We extend our instruction

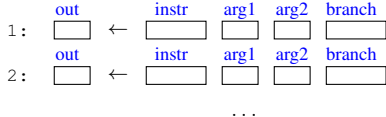

set with jump-if-zero (`jz`), jump-if-not-zero (`jnz`) and `return` instructions. Our assembly program representation also includes a "branch" parameter $b$ specifying the new value of the instruction pointer for a successful conditional branch. To learn programs in this language, the model must learn how to create the control flow that it needs using these simple conditional jumps. Note that the instruction pointer suffers from the same problems as the stack pointer above, i.e., uncertainty about its value blurs together the effects of many possible program executions.

**Structured Control Flow** We see structured control flow as a way to reduce the "bleeding" of uncertainty about the value of the instruction pointer into the values of registers and cells on the heap. To introduce structured control flow, we replace raw jumps with an `if-then-else` instruction and an explicit `foreach` loop that is suited for processing lists. We restrict our model to a prefix of instructions, a loop which iterates over a list, and a suffix of instructions. The parameters for instructions in the loop can access

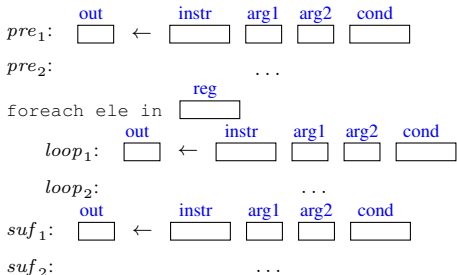

an additional register that contains the value of the current list element. To implement this behavior in practice, we unroll the loop for a fixed number of iterations derived from the bound on the size of the input, which ensures that every input list can be processed. After unrolling, the instruction executed at each timestep becomes deterministic, removing uncertainty about the value of the instruction pointer.

For the `if-then-else` instruction, we extend the instruction representation with a "condition" parameter $c \in [1, R]$ and let the evaluation of `if-then-else` yield its first argument when the register $c$ is non-zero and the second argument otherwise. An overview of the structure of such programs is displayed above.

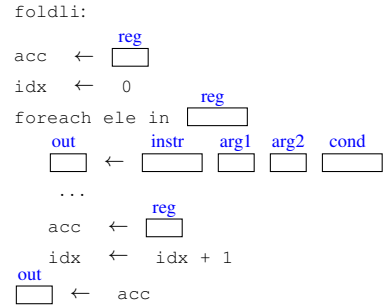

We note that while fixing the iteration over the list elements is already helpful, learning most list-processing programs requires the model to repeatedly infer the concepts of creating a new list, aggregating results and keeping track of the current list index. In functional programming languages, such regular patterns are encapsulated in combinators. Thus, in a second model, we replace the simple `foreach` loop with three combinators: `mapi` creates a new list by applying a function to each element of the input list, `zipWithi` creates a new list by iterating over two lists in parallel and applying a function to both elements, and `foldli` computes a value by iterating over all list elements and applying a function to the current list element and the value computed so far. A program model using the `foldli` combinator is shown on the left. The `i` suffix indicates that these combinators additionally provide the index of the current list element (the precise semantics of the combinators are presented in Sect. A.1).

**Recommendation (L):** Instead of raw jumps, use loop and if-then-else templates.

**Immutable Data** In training our models, we observed that many random initializations of the program parameters would overwrite input data or important intermediate results, and later steps would not be able to recover this data. In models with combinators that provide a way to accumulate result values, we can sidestep this issue by making registers immutable. To do so, we create one register per timestep, and fix the output of each instruction to the register for its timestep. Parameters for arguments then range over all registers initialized in prior timesteps, with an exception for the closures executed by a combinator. Here, each instruction only gets access to the inputs to the closure, values computed in the prefix, and registers initialized by preceding instructions in the same loop iteration. As in the heap allocation case, we can avoid keeping a copy of all registers for every timestep, and instead share these values over all steps, reducing memory usage.

**Recommendation (I):** Use immutable registers by deterministically choosing where to store outputs.

**Types** When training our models, we found that for many initializations, training would fall into local minima corresponding to ill-typed programs, e.g., where references to the heap would be used in integer additions. We expect the learned program to be well-typed, so we introduce a simple type system. We explored two approaches to adding a type system.

A first attempt integrated the well-typedness of the program into our objective function. In our programs, we use three simple types of data—integers, pointers and booleans—as well as a special type, $\perp$, which represents type errors. We extended the program state to contain an additional element $^t r$ for each register, encoding its type. Each instruction then not only computes a value that is assigned to the target register, but also a type for the target register. Most significantly, if one of the arguments has an unsuitable type (e.g., an integer in place of a pointer), the resulting type is $\perp$. We then extended our objective function to add a penalty for values with type $\perp$. Unfortunately, this changed objective function had neither a positive nor negative effect on our experiments, so it seems that optimizing for the correct type is redundant when we are already optimizing for the correct return value.

In our second attempt, rather than penalizing ill-typed programs, we prevent programs from accessing ill-typed data by construction. We augment our register representation by adding an integer, pointer, and Boolean slot to each register, so each register can hold a separate value of each type. Instructions which read from registers now read from the slot corresponding to the type of the argument. When writing to a register, we write to the slot corresponding to the instruction's return type, and set the other slots to a default value 0. This prevents any ill-typed sequence of instructions, i.e., it is now impossible to, for example, increment a pointer value or to construct a cons cell with a non-pointer value. Furthermore, this modification allows us to set the heap size $H$ to a value different from the maximal integer $M$. Our experiments in Sect. 4.3 show that separating differently-typed values simplifies the learning of programs that operate on lists and integers at the same time.

**Recommendation (T):** Use different storage for data of different types.

## 4 EXPERIMENTS

We have empirically evaluated our modeling recommendations on a selection of program induction tasks of increasing complexity, ranging from straight-line programs to problems with loops and conditional expressions. All of our models are implemented in TerpreT (Gaunt et al., 2016b) and we learn using TerpreT's TENSORFLOW (Abadi et al., 2015) backend. We aim to release TerpreT, together with these models, under an open source license in the near future.

For all tasks, three groups of five input/output example pairs were sampled as training data and another 25 input/output pairs as test data. For each group of five examples, training was started from 100 random initializations of the model parameters. After training for 3500 epochs (tests with longer training runs showed no significant changes in the outcomes), the learned programs were tested by discretizing all parameters and comparing program outputs on test inputs with the expected values. We perform 300 runs per model and task, and report only the ratio of successful runs. A run is successful if the discretized program returns the correct result on all five training and 25 test examples.[1] The ratio of runs converging to zero loss on the training examples was within 1% of the number of successful runs, i.e., very few found solutions failed to generalize.

---

[1] We inspected samples of the obtained programs as well and verified that they were indeed correct solutions. See Sect. A.2 for some of the learned programs.

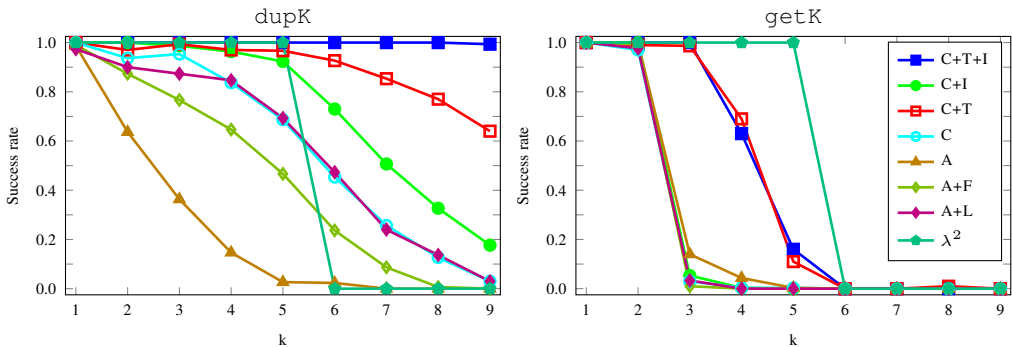

Figure 1: Success rate of our models on straight-line programs of increasing length

We performed a cursory exploration of hyperparameter choices. We varied the choice of optimization algorithm (Momentum, Adam, RMSProp), the learning rate (from 0.001 to 5), gradient noise (testing the recommended choices from Neelakantan et al. (2016b)), a decaying entropy bonus (starting from 0.001 to 20), and gradient clipping (to values between 0.1 and 10). We sampled 100 hyperparameter settings from this space and tested their effect on two simple tasks. We ran the remaining experiments with the best configuration obtained by this process: the RMSProp optimization algorithm, a learning rate of 0.1, clipped gradients at 1, and no gradient noise.

We consider the ratio of successful runs as earlier work has identified this as a significant problem. For example, Neelakantan et al. (2016b) reports that even after a (task-specific) "large grid search" of hyperparameters, the Neural Random Access Machine converged only in 5%, 7% and 22% of random restarts. Similar observations were made in Kaiser & Sutskever (2016); Bunel et al. (2016); Gaunt et al. (2016b) for related program learning models.

In our experiments we evaluate the effect of the choices discussed in Sect. 3, comparing seven model variants in total. We call our initial assembly model A and its variation with a fixed memory allocation scheme A+F. All other models use the same fixed memory allocation scheme. The extension of the assembly model with a built-in `foreach` loop is called A+L. The A+L model also allows a `foreachZip` loop structure that allows parallel iteration over two lists, similar to the `zipWith` combinator. The model including predefined combinators is called C, where C+I (resp. C+T) are its extensions with immutable registers (resp. typed registers). Finally, C+T+I combines all of these and is, in effect, a simple end-to-end differentiable functional programming language.

Additionally, we show results for $\lambda^2$ (Feser et al., 2015), a strong program synthesis baseline from programming languages research, because of its built-in support for list-processing programs. As $\lambda^2$ is deterministic, we only report a success rate of either 1 or 0.

## 4.1 STRAIGHT-LINE PROGRAMS

In our first experiment, we consider two families of simple problems—solvable with straight-line programs—to study the interaction of our modeling choices with program length. Our first benchmark task is to duplicate a scalar input a fixed number $k$ times to create a list of length $k$. Our second benchmark is to retrieve the $k$-th element of a list, again fixing $k$ beforehand (we will consider a generalization of this task where $k$ is a program input later). We set the hyperparameters for all models to allow 11 statements, i.e., for A and A+F we have set the program length to 11, and for the A+L and C* models we have set the prefix and loop length to 0 and the suffix length to 11. For models where the number of registers does not depend on the number of timesteps, we set the number of registers to 3, with one initialized to the input. This allows for $\sim 10^{39}$ programs in the A, A+F, C+I, and C+T+I models and for $\sim 10^{28}$ programs in the remaining models. These parameters were chosen to be slightly larger than required by the largest program to be learned. For all of our experiments, the maximal integer $M$ was set to 20 for models where possible (i.e., for A, C+T+I, C+T), and to $H$ (derived from $T$, coming to 22) for the others.[2]

---

[2]We also experimented with varying the value of $M$. Choices over 20 showed no significant differences to smaller values.

We evaluated all of our models following the regime discussed above and present the results in Fig. 1 for $k$ values from 1 to 9. The difference between A and A+F on the `dupK` task illustrates the significance of **Recommendation (F)** to fix the memory allocation scheme. Following **Recommendation (T)** to separate values of different types improves the results on both tasks, as the differences between C+T+I (resp. C+T) and C+I (resp. C) illustrate.

## 4.2 SIMPLE LOOP PROGRAMS

In our second experiment, we compare our models on three simple list algorithms: computing the length of a list, reversing a list and summing a list. Model parameters have been set to allow 6 statements for the A and A+F models, and empty prefixes, empty suffixes, and 2 instructions in the loop for the other models. For models where the number of registers does not depend on the number of timesteps, we set the number of registers to 4, with one initialized to the input.

| Program | C+T+I | C+T | C+I | C | A | A+F | A+L | $\lambda^2$ |
|---------|-------|-----|-----|---|---|-----|-----|-------------|
| len | **100.00** | 75.00 | **100.00** | 43.67 | 0.00 | 0.00 | 15.67 | 100.00 |
| rev | 48.33 | 32.67 | 46.33 | 41.33 | 0.00 | 0.00 | **86.33** | 100.00 |
| sum | **91.67** | 41.00 | 88.33 | 30.67 | 0.00 | 0.00 | 32.67 | 100.00 |

Table 1: Success ratios for experiments on simple loop-requiring tasks.

The results of our evaluation are displayed in Tab. 1, starkly illustrating **Recommendation (L)** to use predefined loop structures. We speculate that learning explicit jump targets is extremely challenging because changes to the parameters controlling jump target instructions have outsized effects on all computed (intermediate and output) values. On the other hand, models that could choose between different list iteration primitives were able to find programs for all tasks. We again note the effect of **Recommendation (T)** to separate values of different types on the success rates for the `len` and `sum` examples, and the effect of **Recommendation (I)** to avoid mutable data on results for `len` and `rev`.

## 4.3 LOOP PROGRAMS

In our main experiment, we consider a larger set of common list-manipulating tasks (such as checking if all/one element of a list is greater than a bound, retrieving a list element by index, finding the index of a value, and computing the maximum value). Descriptions of all tasks are shown in Fig. 2 in the appendix. We do not show results for the A and A+F models, which always fail. We set the parameters for the remaining models to $M = 32$ where possible ($M = H = 34$ for the others), the length of the prefix to 1, the length of the closure / loop body to 3 and the length of the suffix to 2. Again, these parameters are slightly larger than required by the largest program to be learned.

| Program | C+T+I | C+T | C+I | C | A+L | $\lambda^2$ |
|---------|-------|-----|-----|---|-----|-------------|
| len | **98.67** | 96.33 | 0.67 | 0.33 | 0.00 | 100.00 |
| rev | **18.00** | 10.33 | 2.67 | 8.33 | 9.67 | 100.00 |
| sum | 38.00 | **38.33** | 1.00 | 0.00 | 10.00 | 100.00 |
| allGtK | 0.00 | 0.00 | 0.00 | **0.33** | 0.00 | 100.00 |
| exGtK | **3.00** | 1.00 | 0.67 | 0.00 | 0.67 | 100.00 |
| findLastIdx | **0.33** | 0.00 | 0.00 | 0.00 | 0.00 | 0.00 |
| getIdx | **1.00** | 0.00 | 0.00 | 0.00 | 0.00 | 0.00 |
| last2 | 0.00 | 8.00 | 0.00 | 2.00 | **23.00** | 0.00 |
| mapAddK | **100.00** | 98.00 | **100.00** | 95.67 | 0.00 | 100.00 |
| mapInc | **99.67** | 98.00 | 99.33 | 97.00 | 0.00 | 100.00 |
| max | 2.33 | **5.67** | 0.00 | 0.00 | 0.33 | 100.00 |
| pairwiseSum | 43.33 | 32.33 | **43.67** | 33.67 | 0.00 | 100.00 |
| revMapInc | 0.00 | 0.67 | 0.00 | 0.00 | **6.33** | 100.00 |

Table 2: Success ratios for full set of tasks.

The results for our experiments on these tasks are shown in Tab. 2. Note the changed results of the examples from Sect. 4.2, as the change in model parameters has increased the size of the program space from $\sim 10^7$ to $\sim 10^{20}$. The relative results for the A+L model show the value of built-in iteration and aggregation patterns. The choice between immutable and mutable registers is less clear here, seemingly dampened by other influences. An inspection of the generated programs (eg. Fig. 8 in

the appendix) reveals that mutability of registers can sometimes be exploited to find elegant solutions. Overall, it may be effective to combine both approaches, using a small number of (mutable) "scratch value" registers *and* immutable default output registers for each statement.

## 5 RELATED WORK

**Inductive Program Synthesis**   There has been significant recent interest in synthesizing functional programs from input-output examples in the programming languages community. Synthesis systems generally operate by searching for a program which is correct on the examples, using types or custom deduction rules to eliminate parts of the search space. Among the notable systems: MYTH (Osera & Zdancewic, 2015; Frankle et al., 2016) synthesizes recursive functional programs from examples using types to guide the search for a correct program, $\lambda^2$ (Feser et al., 2015) synthesizes data structure manipulating programs structured using combinators using types and deduction rules in its search, ESCHER (Albarghouthi et al., 2013) synthesizes recursive programs using search and a specialized method for learning conditional expressions, and FlashFill (Gulwani, 2011) structures programs as compositions of functions and uses custom deduction rules to prune candidate programs. Our decision to learn functional programs was strongly inspired by this previous work. In particular, the use of combinators to structure control flow was drawn from Feser et al. (2015). However, our end-to-end differentiable setting is fundamentally different from discrete search employed in the programming languages community, and thus concrete techniques are largely incomparable.

**Neural Networks Learning Algorithms**   A number of recent models aim to learn algorithms from input/output data. Many of these augment standard recurrent neural network architectures with differentiable memory and simple computation components (e.g. Graves et al. (2014); Kurach et al. (2016); Joulin & Mikolov (2015); Neelakantan et al. (2016a); Reed & de Freitas (2016); Zaremba et al. (2016); Graves et al. (2016)). The use of an RNN can be seen as fixed looping structure, and the use of fixed output registers for the modules in Neural Random Access Machines (Kurach et al., 2016) is similar to our modeling of immutable registers.

However, none of these works focus on producing source code. Gaunt et al. (2016b) show that this is an extremely challenging task for assembly-like program models. More recently, Bunel et al. (2016) and Riedel et al. (2016) have used program models similar to assembly (resp. Forth) source code to initialize solutions, and either optimize or complete them.

## 6 DISCUSSION AND FUTURE WORK

We have discussed a range of modeling choices for end-to-end differentiable programming languages and made four design recommendations. Empirically, we have shown these recommendations to significantly improve the success ratio of learning programs from input/output examples, and we expect these results to generalize to similar models attempting to learn programs.

In this paper, we only consider list manipulating programs, but are interested in supporting more data structures, such as arrays (which should be a straightforward extension) and associative maps. We also only support loops over lists at this time, but are interested in extending our models to also have built-in support for loops counting up to (and down from) integer values. A generalization of this concept would be an extension allowing the learning and use of recursive functions. Recursion is still more structured than raw goto calls, but more flexible than the combinators that we currently employ. An efficient implementation of recursion is a challenging research problem, but it could allow significantly more complex programs to be learned. Modeling recursion in an end-to-end differentiable language could allow us to build libraries of (learned) differentiable functions that can be used in later synthesis problems.

However, we note that with few exceptions on long straight-line code, $\lambda^2$ performs better than all of our considered models, and is able to synthesize programs in milliseconds. We see the future of differentiable programming languages in areas in which deterministic tools are known to perform poorly, such as the integration of perceptual data, priors and "soft" side information such as natural language hints about the desired functionality. Gaunt et al. (2016a) was developed in parallel to this work and builds on many of our results to learn programs that can process perceptual data (in the current example, images).

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

# A APPENDIX

| Name | Description |
|------|-------------|
| len | Return the length of a list. |
| rev | Reverse a list. |
| sum | Sum all elements of a list. |
| allGtK | Check if all elements of a list are greater than $k$. |
| exGtK | Check if at least one element of a list is greater $k$. |
| findLastIdx | Find the index of the last list element which is equal to $v$. |
| getIdx | Return the $k$th element of a list. |
| last2 | Return the 2nd to last element of a list. |
| mapAddK | Compute list in which $k$ has been added to each element of the input list. |
| mapInc | Compute list in which each element of the input list has been incremented. |
| max | Return the maximum element of a list. |
| pairwiseSum | Compute list where each element is the sum of the corresponding elements of two input lists. |
| revMapInc | Reverse a list and increment each element. |

Figure 2: Our example tasks for loop based programs. "Simple" tasks are above the line.

## A.1 COMBINATORS

**function** FOLDLI($list, acc, func$)
 $idx \leftarrow 0$
 **for** $ele$ **in** $list$ **do**
 $acc \leftarrow func(acc, ele, idx)$
 $idx \leftarrow idx + 1$
 **return** $acc$

**function** ZIPWITHI($list_1, list_2, func$)
 $idx \leftarrow 0$
 $ret \leftarrow [\,]$
 **for** $ele_1, ele_2$ **in** $list_1, list_2$ **do**
 $ret \leftarrow append(ret, func(ele_1, ele_2, idx))$
 $idx \leftarrow idx + 1$
 **return** $ret$

**function** MAPI($list, func$)
 $idx \leftarrow 0$
 $ret \leftarrow [\,]$
 **for** $ele$ **in** $list$ **do**
 $ret \leftarrow append(ret, func(ele, idx))$
 $idx \leftarrow idx + 1$
 **return** $ret$

Figure 3: Semantics of foldli, mapi, zipwithi in a Python-like language.

## A.2 SELECTED SOLUTIONS

We show example results of our training in Figs. 4-16. Note that these are the actual results produced by our system, and have only been slightly edited for typesetting. Finally, we have colored statements that a simple program analysis can identify as not contributing to the result in gray.

**let** $r_0 = l$ **in**
**let** $r_1 = k$ **in**
let $r_2 = (r_0 = r_1)$ in
**let** $r_3 = $ foldli $r_0 \, r_0 \, (\lambda \, ele \, acc \, idx \rightarrow$
 **let** $c_0 = acc \vee acc$ **in**
 **let** $c_1 = ele > r_1$ **in**
 **let** $c_2 = c_0 \vee c_1$ **in**
 $c_2$) **in**
let $r_4 = r_3 \vee r_3$ in
let $r_5 = r_3 \wedge r_2$ in
**return** $r_4$

$r_0 \leftarrow l$
$r_1 \leftarrow k$
$r_2 \leftarrow r_2 = r_1$
**for** $ele$ **in** $r_0$ **do**
 $r_0 \leftarrow$ **if** $r_2$ **then** $ele$ **else** $r_1$
 $r_0 \leftarrow ele > r_1$
 $r_2 \leftarrow r_2 \vee r_0$
$r_2 \leftarrow r_2 \vee r_0$
$r_1 \leftarrow r_2 \vee r_2$
**return** $r_2$

Figure 5: Solutions to `exGtK` in the C+T+I and A+L models.

**let** $r_0 = l$ **in**
**let** $r_1 = e$ **in**
**let** $r_2 = r_0 + 1$ **in**
**let** $r_3 = $ foldli $r_0 \, r_2 \, (\lambda \, ele \, acc \, idx \rightarrow$
 let $c_0 = $ **if** $r_2$ **then** $idx$ **else** $r_1$ in
 **let** $c_1 = (r_1 = ele)$ **in**
 **let** $c_2 = $ **if** $c_1$ **then** $idx$ **else** $acc$ **in**
 $c_2$) **in**
let $r_4 = r_3 + 1$ in
let $r_5 = r_2$ in
**return** $r_3$

Figure 6: A solution to `findLastIdx` in the C+T+I model.

$r_0 \leftarrow l$
$r_1 \leftarrow k$
$r_2 \leftarrow r_0 \vee r_0$
$r_1 \leftarrow$ foldli $r_0 \, r_0 \, (\lambda \, ele \, acc \, idx \rightarrow$
 $r_0 \leftarrow ele > r_1$
 $r_2 \leftarrow$ car $acc$
 $r_2 \leftarrow r_0 \wedge acc$
 $r_2$)
$r_2 \leftarrow r_1 \wedge r_0$
$r_1 \leftarrow r_1$
**return** $r_2$

Figure 4: A solution to `allGtK` in the C model. Code in gray is dead.

**let** $r_0 = l$ **in**
**let** $r_1 = k$ **in**
**let** $r_2 = $ car $r_0$ **in**
**let** $r_3 = $ foldli $r_0 \, r_2 \, (\lambda \, ele \, acc \, idx \rightarrow$
 **let** $c_0 = (r_1 = idx)$ **in**
 **let** $c_1 = $ **if** $c_0$ **then** $ele$ **else** $acc$ **in**
 let $c_2 = $ **if** $idx$ **then** $idx$ **else** $c_0$ in
 $c_1$) **in**
let $r_4 = $ car $r_0$ in
let $r_5 = $ cdr $r_0$ in
**return** $r_3$

Figure 7: A solution to `getIdx` in the C+T+I model.

```
r_0 ← l
r_1 ← 0
r_2 ← nil
r_2 ← foldli r_0 r_1 (λ ele acc idx →
    r_0 ← acc
    r_2 ← r_1
    r_1 ← ele
    r_2) in
r_0 ← r_2 + r_2
r_0 ← r_0 + 1
return r_2
```

```
r_0 ← l
r_1 ← 0
r_2 ← cons r_0 r_0
for (ele_1, ele_2) in (r_0, r_2) do
    r_2 ← if r_2 then ele_2 else ele_2
    r_1 ← r_2 − 1
    r_1 ← car r_0
r_1 ← if r_1 then r_2 else r_0
r_0 ← if r_2 then r_0 else r_1
return r_2
```

Figure 8: Solutions to `last2` in the C+T and A+L models.

```
let r_0 = l in
let r_1 = 0 in
let r_2 = cdr r_0 in
let r_3 = foldli r_0 r_0 (λ ele acc idx →
    let c_0 = idx + 1 in
    let c_1 = if r_1 then r_2 else c_0 in
    let c_2 = c_0 = ele in
    c_0) in
let r_4 = if r_2 then r_3 else r_3 in
let r_5 = r_3 + r_2 in
return r_3
```

Figure 9: A solution to `len` in the C+T+I model.

```
let r_0 = l in
let r_1 = k in
let r_2 = if r_1 then r_0 else r_0 in
let r_3 = mapi r_0 (λ ele idx →
    let c_0 = ele − 1 in
    let c_1 = c_0 − 1 in
    let c_2 = r_1 + ele in
    c_2) in
let r_4 = r_3 in
let r_5 = r_3 in
return r_3
```

Figure 10: A solution to `mapAddK` in the C+T+I model.

```
let r_0 = l in
let r_1 = 0 in
let r_2 = r_0 in
let r_3 = mapi r_0 (λ ele idx →
    let c_0 = if r_1 then ele else acc in
    let c_1 = ele + 1 in
    let c_2 = r_1 in
    c_1) in
let r_4 = cons r_3 r_0 in
let r_5 = r_4 in
return r_3
```

Figure 11: A solution to `mapInc` in the C+T+I model.

```
let r0 = l in
let r1 = 0 in
let r2 = cdr r0 in
let r3 = foldli r0 r0 (λ ele acc idx →
    let c0 = acc > ele in
    let c1 = acc in
    let c2 = if c0 then acc else ele in
    c2) in
let r4 = r2 − 1 in
let r5 = car r2 in
return r3
```

```
r0 ← l
r1 ← 0
r2 ← r0 − 1
for (ele1, ele2) in (r0, r0) do
    r0 ← ele1
    r1 ← ele1 > r2
    r2 ← if r1 then ele1 else r2
r0 ← r0 + r0
r0 ← cons r2 r2
return r2
```

Figure 12: Solutions to max in the C+T+I and A+L models.

```
let r0 = l1 in
let r1 = l2 in
let r2 = if r1 then r0 else r1 in
let r3 = zipWithi r1 r0 (λ ele1 ele2 idx →
    let c0 = ele1 + ele2 in
    let c1 = ele2 − 1 in
    let c2 = idx − 1 in
    c0) in
let r4 = if r0 then r3 else r1 in
let r5 = if r4 then r2 else r1 in
return r3
```

Figure 13: A solution to pairwiseSum in the C+T+I model.

```
let r0 = l in
let r1 = 0 in
let r2 = cons r0 r0 in
let r3 = foldli r0 r1 (λ ele acc idx →
    let c0 = cons ele acc in
    let c1 = cons acc acc in
    let c2 = cons ele acc in
    c2) in
let r4 = if r2 then r3 else r2 in
let r5 = cons r4 r3 in
return r3
```

```
r0 ← l
r1 ← 0
r2 ← cdr r1
for ele1 in r0 do
    r1 ← cons ele2 r0
    r2 ← cons ele1 r2
    r1 ← cdr r0
r0 ← cdr r0
r0 ← cons r2 r1
return r2
```

Figure 14: Solutions to rev in the C+T+I and A+L models.

```
r0 ← l
r1 ← 0
r1 ← cdr r0
r2 ← foldli r0 r2 (λ ele acc idx →
    r2 ← ele + 1
    r0 ← cons r2 acc
    r2 ← cons r2 acc
    r2)
r1 ← cons r2 r1
r0 ← cons r2 r0
return r2
```

```
r0 ← l
r1 ← 0
r1 ← 1
for ele1 in r0 do
    r0 ← ele1 + 1
    r1 ← 1
    r2 ← cons r0 r2
r1 ← cons r0 r2
r1 ← cons r0 r2
return r2
```

Figure 15: Solutions to revMapInc in the C+T and A+L models.

**let** $r_0 = l$ **in**
**let** $r_1 = 0$ **in**
**let** $r_2 = r_0$ **in**
**let** $r_3 = $ `foldli` $r_0$ $r_0$ $(\lambda\, ele\, acc\, idx \rightarrow$
      **let** $c_0 = acc + r_0$ **in**
      **let** $c_1 = acc + ele$ **in**
      **let** $c_2 = $ **if** $r_0$ **then** $idx$ **else** $r_1$ **in**
      $c_1)$ **in**
**let** $r_4 = r_2 + 1$ **in**
**let** $r_5 = r_3 - 1$ **in**
**return** $r_3$

$r_0 \leftarrow l$
$r_1 \leftarrow 0$
$r_1 \leftarrow$ **if** $r_2$ **then** $r_1$ **else** $r_0$
**for** $ele_1$ **in** $r_0$ **do**
      $r_2 \leftarrow ele_1 + r_2$
      $r_1 \leftarrow$ `cons` $r_2$ $r_0$
      $r_1 \leftarrow ele_1 + ele_2$
$r_0 \leftarrow r_1 + r_1$
$r_0 \leftarrow r_2 + 1$
**return** $r_2$

Figure 16: Solutions to `sum` in the C+T+I and A+L models.

