# Peer review of "Neural Functional Programming"

_ICLR 2017 — rejected_

[Official Review · AnonReviewer1 · rating 6 · confidence 3 · 15 Dec 2016]
**Weak accept**

This paper presents small but important modifications which can be made to differentiable programs to improve learning on them. Overall these modifications seem to substantially improve convergence of the optimization problems involved in learning programs by gradient descent. That said, the set of programs which can be learned is still small, and unlikely to be directly useful.

[Official Review · AnonReviewer3 · rating 5 · confidence 2 · 16 Dec 2016]
**Not sure if very interesting to the ICLR community**

The paper proposes a set of recommendations for the design of differentiable programming languages, based on what made gradient descent more successful in experiments.

I must say i’m no expert in program induction. While i understand there is value in exploring what the paper set out to explore -- making program learning easier -- i did not find the paper too engaging. First everything is built on top of Terpret, which isn’t yet publicly available. Also most of the discussion is very detailed on the programming language side and less so on the learning side. It is conceivable that it would be best received on a programming language conference. A comparison with alternatives not generating code would be valuable in my opinion, to motivate for the overall setup.

Pros: 
Useful, well executed, novel study.
Cons:
Low on learning-specific contributions, more into domain-related constraints. Not sure a great fit to ICLR.

[Official Review · AnonReviewer4 · rating 4 · confidence 3 · 17 Dec 2016]

The paper discusses a range of modelling choices for designing differentiable programming languages. Authors propose 4 recommendations that are then tested on a set of 13 algorithmic tasks for lists, such as "length of the list", "return k-th element from the list", etc. The solutions are learnt from input/output example pairs (5 for training, 25 for test).

The main difference between this work and differentiable architectures, like NTM, Neural GPU, NRAM, etc. is the fact that here the authors aim at automatically producing code that solves the given task.

My main concern are experiments - it would be nice to see a comparison to some of the neural networks mentioned in related work. Also, it would be useful to see how this model is doing on typical problems used by mentioned neural architectures (problems such as "sorting", "merging", "adding"). I'm wondering how this is going to generalize to other types of programs that can't be solved with prefix-loop-suffix structure.

It is also concerning that although  1) the tasks are simple, 2) the structure of the solution is very restricted and 3) model is using extensions doing most of the work, the proposed model still fails to find solutions (example: A+L model that has “loop” fails to solve “list length” task in 84% of the runs).


Pro:
- generates code rather than black-box neural architecture
- nice that it can learn from very few examples

Cons:
- weak results, works only for very simple tasks, missing comparison to neural architectures

[Official Review · AnonReviewer2 · rating 7 · confidence 2 · 17 Dec 2016]

This paper presents design decisions of TerpreT [1] and experiments about learning simple loop programs and list manipulation tasks. The TerpreT line of work (is one of those which) bridges the gap between the programming languages (PL) and machine learning (ML) communities. Contrasted to the recent interest of the ML community for program induction, the focus here is on using the design of the programming language to reduce the search space. Namely, here, they used the structure of the control flow (if-then-else, foreach, zipWithi, and foldli "templates"), immutable data (no reuse of a "neural" memory), and types (they tried penalizing ill-typedness, and restricting the search only to well-typed programs, which works better). My bird eye view would be that this stands in between "make everything continuous and perform gradient descent" (ML) and "discretize all the things and perform structured and heuristics-guided combinatorial search" (PL).

I liked that they have a relevant baseline (\lambda^2), but I wished that they also included a fully neural network program synthesis baseline. Admittedly, it would not succeed except on the simplest tasks, but I think some of their experimental tasks are simple enough for "non-generating code " NNs to succeed on.

I wished that TerpreT was available, and the code to reproduce these experiments too.

I wonder if/how the (otherwise very interesting!) recommendations for the design of programming languages to perform gradient descent based-inductive programming would hold/perform on harder task than these loops. Even though these tasks are already interesting and challenging, I wonder how much of these tasks biased the search for good subset of constraints (e.g. those for structuring the control flow).

Overall, I think that the paper is good enough to appear at ICLR, but I am no expert in program induction / synthesis.

Writing:
 - The paper is at times hard to follow. For instance, the naming scheme of the model variants could be summarized in a table (with boolean information about the features it embeds).
 - Introduction: "basis modern computing" -> of
 - Page 3, training objective: "minimize the cross-entropy between the distribution in the output register r_R^{(T)} and a point distribution with all probability mass on the correct output value" -> if you want to cater to the ML community at large, I think that it is better to say that you treat the output of r_R^{(T)} as a classification problem with the correct output value (you can give details and say exactly which type of criterion/loss, cross-entropy, you use).


[1] "TerpreT: A Probabilistic Programming Language for Program Induction", Gaunt et al. 2016

[Official Review · AnonReviewer5 · rating 5 · confidence 3 · 20 Dec 2016]
**very interesting but probably too derivative from earlier already published work**

The authors talk about design choice recommendations for performing program induction via gradient descent, basically advocating reasonable programming language practice (immutable data, higher-order language constructs, etc.).  

As mentioned in the comments I feel fairly strongly that this is a marginal at best contribution beyond TerpreT, an already published system with extensive experimentation and theoretical grounding.  To be clear I think the TerpreT paper deserves a large amount of attention.  It is truly inspiring.

This paper contradicts one of the key findings in the original paper but doesn't provide convincing evidence that gradient-based evaluators for TerpreT are superior or even, frankly, appropriate for program induction.   This is uncomfortable for me and makes me wonder why gradient-based methods weren't more carefully vetted in the first place or why more extensive comparisons to already implemented alternatives weren't included in this paper.  

My opinion: if we want to give the original TerpreT paper more attention, which I think it deserves, then this paper is above threshold.  On the other hand it's basically unreadable, actually contradicts its mother-paper in not well-defended ways, and is irreproducible without the same so I think, unfortunately, it's below threshold.

[Author Response · Marc Brockschmidt · 17 Jan 2017]
**General response**

Thank you to all the reviewers for their comments. We believe that these
are the primary points raised by the reviewers:

1. A comparison with a neural programming technique which does not
   generate code would be a valuable addition to our experiments.

   We did not include a comparison to a network which does not
   generate source code because because they usually require
   substantially more training data than the 5 input-output examples
   we provide to have any success at generalization.
   While it would be interesting to show this effect in experiments,
   the wide variety of different models and training strategies,
   the custom structure of list data and list-aware objective function in our work,
   together with the lack of released standard implementations,
   makes it unclear what neural programming baseline (and with what
   training regime) would be appropriate.

2. The tasks our network can learn are simple. In particular, we do
   not consider sorting or merging problems.

   Although the tasks that we considered are simple, they are more complex
   than the tasks which many other neural programming approaches can handle,
   particularly as we test for perfect generalization.

   The approaches to learning to sort do so using program traces,
   which provide much stronger supervision than the input-output examples
   that we use [1], or use specialized memory representations that simplify
   sorting [2]. 

3. This paper does not conclusively show that gradient-based
   evaluators are appropriate for program induction.

   This paper certainly does not contradict the findings of the
   original TerpreT paper that discrete solvers are good backends for
   program induction. Our motivation in this paper is to improve gradient-based
   program search, and to understand the effect of different design choices
   that arise when building differentiable interpreters. Even if gradient descent
   isn't currently the best method for program induction, we believe it merits
   further study. It is a very new idea, and it's feasible to us that seemingly
   subtle design decisions could make a big difference in its performance (indeed,
   this is one take-away from our experiments). Further, since gradient descent
   is very different from alternatives, improving its performance may enable 
   new uses of program synthesis such as jointly inducing programs and training
   neural network subcomponents as in [3], using SGD to scale up to large data
   sets, or giving new ways of thinking about noisy data in program synthesis.

   Finally, the recommendations for the design of such evaluators
   discussed in this paper are not necessarily restricted to TerpreT-based models,
   and we believe that our design recommendations apply to related
   neural architectures that try to learn algorithmic patterns.

4. How will this model generalize to programs that can't be solved
   using the prefix-loop-suffix structure?

   Defining new program structures is simple, and our current
   implementation allows the optimizer to choose between several
   program structures. More loops could be added if desired, and more
   looping schemes could be added to extend the class of programs
   which can be learned.

5. TerpreT is not yet publicly available.

   TerpreT is nearly ready for public release. Approval for open-sourcing
   under the MIT license has been obtained, and we are currently in the
   process of documenting the source code to publish it (with all models
   used in this paper).


References:
[1] Scott Reed and Nando de Freitas. Neural Programmer-Interpreters.
    In ICLR 2016.
[2] Marcin Andrychowicz, Karol Kurach. Learning Efficient Algorithms
    with Hierarchical Attentive Memory.

[Final Decision · Program Chairs · 06 Feb 2017]
**ICLR committee final decision**

Quality, Clarity: There is no consensus on this, with the readers having varying backgrounds, and one reviewer commenting that they found it to be unreadable. 
 
 Originality, Significance:
  The reviews are mixed on this, with the high score (7) acknowledging a lack of expertise on program induction.
 The paper is based on the published TerpreT system, and some think that it marginal and contradictory with respect to the TerpreT paper. In the rebuttal, point (3) from the authors points to the need to better understand gradient-based program search, even if it is not always better. This leaves me torn about a decision on this paper, although currently it does not have strong support from the most knowledgeable reviewers.
 That said, due to the originality of this work, the PCs are inclined to invite this work to be presented as a workshop contribution.